# Comparing the effects of dynamic and holding isometric contractions on cardiovascular, perceptual, and near-infrared spectroscopy parameters: A pilot study

Daniel Santarém[1]*, Isabel Machado[1,2], Jaime Sampaio[1,2], Catarina Abrantes[1,2]

**1** Department of Sports Science, Exercise and Health, University of Trás-os-Montes and Alto Douro (UTAD), Vila Real, Portugal, **2** Research Center in Sports Sciences, Health Sciences and Human Development, CIDESD, UTAD, Vila Real, Portugal

* danielrs@utad.pt

**Data Availability Statement:** All relevant data are within the paper.

## Abstract

The aim of this pilot study was to assess the effect of muscle contraction type on $SmO_2$ during a dynamic contraction protocol (DYN) and a holding isometric contraction protocol (ISO) in the back squat exercise. Ten voluntary participants (age: 26.6 ± 5.0 years, height: 176.8 ± 8.0 cm, body mass: 76.7 ± 8.1 kg, and one-repetition maximum (1RM): 112.0 ± 33.1 kg) with back squat experience were recruited. The DYN consisted of 3 sets of 16 repetitions at 50% of 1RM (56.0 ± 17.4 kg), with a 120-second rest interval between sets and 2 seconds per movement cycle. The ISO consisted of 3 sets of 1 isometric contraction with the same weight and duration as the DYN (32 seconds). Through near-infrared spectroscopy (NIRS) in the *vastus lateralis* (VL), *soleus* (SL), *longissimus* (LG), and *semitendinosus* (ST) muscles, the minimum $SmO_2$ ($SmO_{2\ min}$), mean $SmO_2$ ($SmO_{2\ avg}$), percent change from baseline ($SmO_{2\ \Delta deoxy}$) and time to recovery 50% of baseline value ($t\,SmO_{2\ 50\%reoxy}$) were determined. No changes in $SmO_{2\ avg}$ were found in the VL, LG, and ST muscles, however the SL muscle had lower values in DYN, in the 1st set ($p = 0.002$) and in the 2nd set ($p = 0.044$). In terms of $SmO_{2\ min}$ and $\Delta SmO_{2\ deoxy}$, only the SL muscle showed differences ($p \leq 0.05$) and lower values in the DYN compared to ISO regardless of the set. The $t\,SmO_{2\ 50\%reoxy}$ was higher in the VL muscle after ISO, only in the 3rd set. These preliminary data suggested that varying the type of muscle contraction in back squat with the same load and exercise time resulted in a lower $SmO_{2\ min}$ in the SL muscle in DYN, most likely because of a higher demand for specialized muscle activation, indicating a larger oxygen supply-consumption gap.

## Introduction

The back squat is one of the most popular exercises in training sessions, being mostly constrained by the differences in body types, leg length, and ankle mobility [1]. During this exercise, the *vastus lateralis* (VL) muscle act as primary mover, *longissimus* (LG) and *semitendinosus* (ST) muscles act as stabilizers, and *soleus* (SL) muscle act as secondary capacity [2–4].

**Funding:** HEALTH-UNORTE: Setting-up biobanks and regenerative medicine strategies to boost research in cardiovascular, musculoskeletal, neurological, oncological, immunological and infectious diseases (NORTE-01-0145-FEDER-000039), financed by Fundo Europeu de Desenvolvimento Regional (FEDER) by NORTE 2020 (Programa Operacional Regional do Norte 2014/2020).

**Competing interests:** The authors have declared that no competing interests exist.

Differences in muscle activity during dynamic and isometric squat exercise have been poorly studied [5]. Concerning the mechanical action of the muscles, while dynamic exercise is characterised by changes in skeletal muscle length and joint movement with rhythmic contractions that raise a relatively small intramuscular force, the isometric exercise induces a relatively large intramuscular force with little or no change in skeletal muscle length or joint movement [6]. On the one hand, dynamic contractions are defined by concentric and eccentric muscle actions, with a relatively easy differentiation. On the other hand, isometric contractions can also take two forms, as holding muscle action, related to holding an inertial load, and pushing isometric muscle action, related to pushing against a stable resistance [7]. Of these two modes of isometric manifestation, holding muscle actions are the easier to apply and evaluate, however, both are rarely the subject of research study in this area. Moreover, as the dynamic strength exercise is considered the most favourable exercise mode for strength gains that will later positively influence sports related to dynamic performance [8], the isometric strength training is considered a feasible alternative mode of training that induces less fatigue, superior angle specific strength and benefit various sports related to dynamic performance [9]. Since dynamic contraction exercises are most frequently included in resistance training programs in different populations [10, 11], isometric training enables a precisely regulated application of force within pain-free joint angles, with application in different clinical and training scenarios.

Monitoring training is becoming indispensable to fine-tune the dose-response and, ultimately, improving performance. Muscle oxygen saturation ($SmO_2$) has been gaining emphasis as a local muscle measurement at rest and during exercise [12], not only in terms of sports performance [13] but also in terms of health [14, 15]. Near-infrared spectroscopy (NIRS) is a non-invasive method that continuously monitors information about the changes in oxygenation and haemodynamics in muscle tissue [16]. The $SmO_2$ reflects the dynamic balance between oxygen supply and oxygen consumption in the examined muscle [17]. $SmO_2$-derived parameters such as percentage deoxygenation ($\Delta SmO_{2\ deoxy}$) and reoxygenation time to 50% ($t\ SmO_{2\ 50\%reoxy}$) may be critical aspects in training planning and monitoring, where a higher $\Delta SmO_{2\ deoxy}$ and a shorter $t\ SmO_{2\ 50\%reoxy}$ time may be associated with better performance. During exercise, $SmO_2$ kinetics can be different depending on several factors, including velocity and intensity of contraction [18], muscle fascicle length and fascicle angle [19], and type of fiber present in the muscle [20]. However, the effect of type of muscle contraction (dynamic and isometric) has been barely explored.

In order to solve the lack of information during resistance training and more precisely the influence of different types of muscle contraction and its effects on the balance between oxygen supply and consumption, the aim of this study was to compare the variations in $SmO_2$ between dynamic and holding isometric contractions in the back squat exercise. As a secondary objective, we investigated the effect of muscular contraction type along the 3 sets. Thus, it was hypothesized that different types of muscle contractions would induce distinct responses in $SmO_2$.

## Material and methods

### Participants

Ten participants (age: 26.6 ± 5.0 years; body mass: 76.7 ± 8.1 kg; body height: 176.8 ± 8.0 cm) volunteered to participate in this study (Table 1) and met the following inclusion criteria: i) familiarization with back squat exercise; ii) physically active, according to the recommendations of the World Health Organization; iii) without musculoskeletal injuries that could affect the protocol procedures; and iv) apparently healthy. Exclusion criteria included: i) lower limb

**Table 1. Physical and physiological characteristics of the participants (n = 10).**

| Variable | Mean ± standard deviation |
|---|---|
| Age (years) | 26.6 ± 5.0 |
| Body height (cm) | 176.8 ± 8.0 |
| Body weight (kg) | 76.7 ± 8.1 |
| VL skinfold (mm) | 10.30 ± 3.62 |
| SL skinfold (mm) | 8.90 ± 3.60 |
| ST skinfold (mm) | 6.50 ± 1.96 |
| LG skinfold (mm) | 8.90 ± 3.31 |
| 1RM (kg) | 112.0 ± 33.1 |
| HR rest (bpm) | 72.8 ± 10.3 |
| DBP rest (mmHg · bpm) | 116.0 ± 8.1 |
| SBP rest (mmHg · bpm) | 72.5 ± 7.7 |

The values are mean ± standard deviation. VL, *vastus lateralis*; SL, *soleus*; ST, *semitendinosus*; LG, *longissimus*; 1RM, one-repetition maximum; HR, heart rate; DBP, diastolic blood pressure; SBP, systolic blood pressure.

injuries in the last year; and ii) anterior lumbar back injury. None of them had any history or clinical signs of cardiovascular or pulmonary disease. Skinfold thickness was measured at the sites of placement of the NIRS devices, using a skinfold caliper (Slim Guide, EUA), to ensure that skinfold thickness was less than 15 mm [21]. In addition, only Caucasians were selected because melanin skin can affect the signal strength of NIRS technology [12]. Before the study started, all participants were informed about the study procedures, provided written informed consent, and completed the Physical Activity Readiness Questionnaire. The protocol was approved by the ethics committee of the University of Trás-os-Montes and Alto Douro (Doc94-CE-UTAD-2021), in accordance with the Declaration of Helsinki.

## Test procedure

This research was conducted in a sports physiology laboratory, under controlled environmental conditions. The participants were instructed not to perform moderate-vigorous intensity physical activity during the 24h before the experiment. They were advised to avoid ingesting alcohol, caffeine, tobacco, or other stimulants and food 3h prior to the test.

Each participant went to the lab on three occasions with at least a 48-h interval between sessions, with all testing procedures performed by the same researcher. During the first session, the participants were clarified about the experimental procedures, signed the informed consent, and made a familiarisation with the exercise protocol and one-repetition maximum (1RM) test. In the second session 1RM in the back squat was determined according to Kraemer and Fry's methodology [22]. In the third session, the two exercise protocols were randomly performed. Before the first protocol, a 10-minute rest in a seated position with back support and feet on the floor was taken and a standardized warm-up was performed (12 alternating knee elevations, 12 alternating knee flexions, 10 jumping jacks, 10 dynamic squats, 12 alternating lunges, 1 isometric squat with a duration of 10 seconds, 12 alternating lunges and 2 minutes of free warm-up). After that, a rest time of 5 minutes has been provided. Between protocols, participants take 10 minutes of passive rest, then a re-warm-up was conducted (10 dynamic squats, 12 alternating lunges, 1 isometric squat with a duration of 10 seconds, and 12 alternating lunges) and retake a seated position for 2 minutes to stabilise the physiological variables.

The dynamic contractions protocol (DYN) consisted of 3 sets of 16 repetitions at 50% of 1RM. The cadence was set at 60 beats per minute using a digital metronome at a tempo of 1

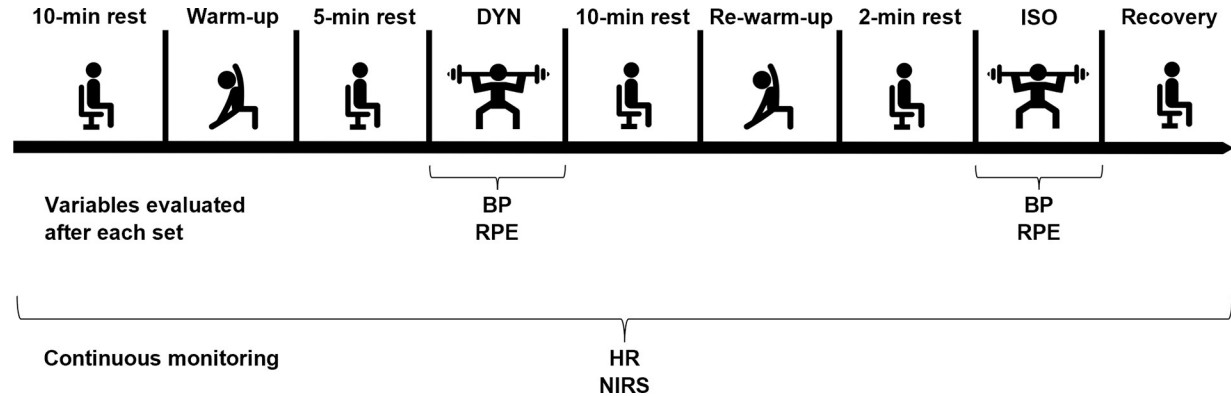

**Fig 1. Schematic representation of the experimental procedures.** DYN, dynamic contractions protocol; ISO, isometric contractions protocol; BP, blood pressure; RPE, rate of perceived exertion; HR, heart rate; NIRS, near-infrared spectroscopy.

second for both concentric and eccentric contractions with a 120-second rest interval between sets. The participants assumed an initial stance position with feet placed approximately shoulder-width apart and the bar placed on the *trapezius* muscle (high-bar position). The squat movement started from an upright position, with knees and hips fully extended. Then, they squatted down until the knee angle was 90˚ measured with a digital goniometer (Halo, Daviscomms, Techpark, Singapura) placed at the knee joint, and returned to the initial position. Thus, to increase the consistency of the squat, an elastic band was hung at 90˚ of knee flexion so that participants know when the descent phase ended, and the ascent phase started. In the isometric contractions protocol (ISO), participants performed 3 sets of 1 isometric contraction at 50% of 1RM. The isometric contraction time was the same as DYN, corresponding to 32 seconds. The critical components for the correct movement execution were the same as DYN, with the particularity that the participants were instructed to adopt a knee angle of 90˚, confirmed before every set using the goniometer. Moreover, the elastic band was also placed at that angle. For both protocols, each set was visually monitored, and verbal instructions were transmitted to ensure proper technique.

The protocol procedures of the evaluation sessions are represented in Fig 1.

**Muscle oxygen saturation measurement.** During the tests, data was collected continuously using a validated and reliable portable wireless NIRS device (MOXY, Fortiori Design LLC, Hutchinson, MN, USA), which applies continuous light from near-infrared wavelength spectrum (light from about 680–810 nm). The distance between the emitter and the two detectors is 12.5 and 25.0 mm. With resource to both Beer-Lambert Law and spatial resolution method, the Moxy Monitor estimates $SmO_2$ and total hemoglobin (tHb) levels in muscle capillaries below its point of position. By default, the NIRS data is acquired with a frequency of 0.5 Hz. Four NIRS sensors were placed on the muscles on the dominant side of the participants: in LG, at 2 finger width lateral from the spinous process of L1; in SL, at 2/3 of the line between the medial condyle of the femur to the medial malleolus; in ST, at 50% on the line between the ischial tuberosity and the medial epicondyle of the tibia; and in VL, at 2/3 on the line from the anterior superior iliac spinae to the lateral side of the patella, according to the SENIAM project for electromyography measurements [23] and were marked with a permanent marker to record and replicate for the consequent sessions. The emitter and detectors were placed parallel to the direction of muscle fibers. To attach and protect from environmental light intrusion, the NIRS sensors were fixed with the material suggested by the manufacturer and an athletic tape.

The NIRS devices and heart rate belt were connected to a computer via ANT+ technology for data visualization, with the use of SPro software (RealTrack Systems, Almería, Spain). An inertial device WIMU PRO (RealTrack Systems, Almería, Spain) was used to synchronise the data from NIRS devices and the heart rate belt.

**Muscle oxygen saturation parameters assessment.** $SmO_2$ parameters were determined through SPro software and Microsoft Excel for Windows, and are presented in Fig 2.

$SmO_{2\ baseline}$ was calculated using the average of the last 20 seconds preceding the exercise. The minimum $SmO_2$ value ($SmO_{2\ min}$) was defined as the minimum value achieved after the implementation of the stimulus. $SmO_2$ average ($SmO_{2\_avg}$) represented the average value during the set. $\Delta SmO_{2\ deoxy}$ was set as the difference between baseline $SmO_2$ and $SmO_{2\ min}$. $t\ SmO_{2\ 50\%reoxy}$ was defined as the time from $SmO_{2\_min}$ up to 50% of the baseline value. The minimum tHb value ($tHb_{min}$) was defined as the minimum value achieved after the implementation of the stimulus, and the average tHb ($tHb_{avg}$) represented the average value during the set.

**Blood pressure (BP).** BP was measured once, immediately after each set, using an electronic blood pressure monitor (Omron 705IT, Healthcare CO., Ukyoku, Kioto, Japan). This measurement was done in a seated position with the back supported, and with the left arm supported at heart level. Mean arterial pressure (MAP) is defined as the average arterial pressure over one cardiac cycle, systole, and diastole, and was calculated using the following formula: MAP = ((2(DBP) +SBP)/3, where DBP represents the diastolic blood pressure and SBP the systolic blood pressure.

**Heart rate (HR).** HR was monitored continuously from a heart rate belt (Garmin, Soft Strap Premium, Lenetsa, KS, USA). Rate pressure product (RPP) was calculated by multiplying the values for HR and SBP measured after each set.

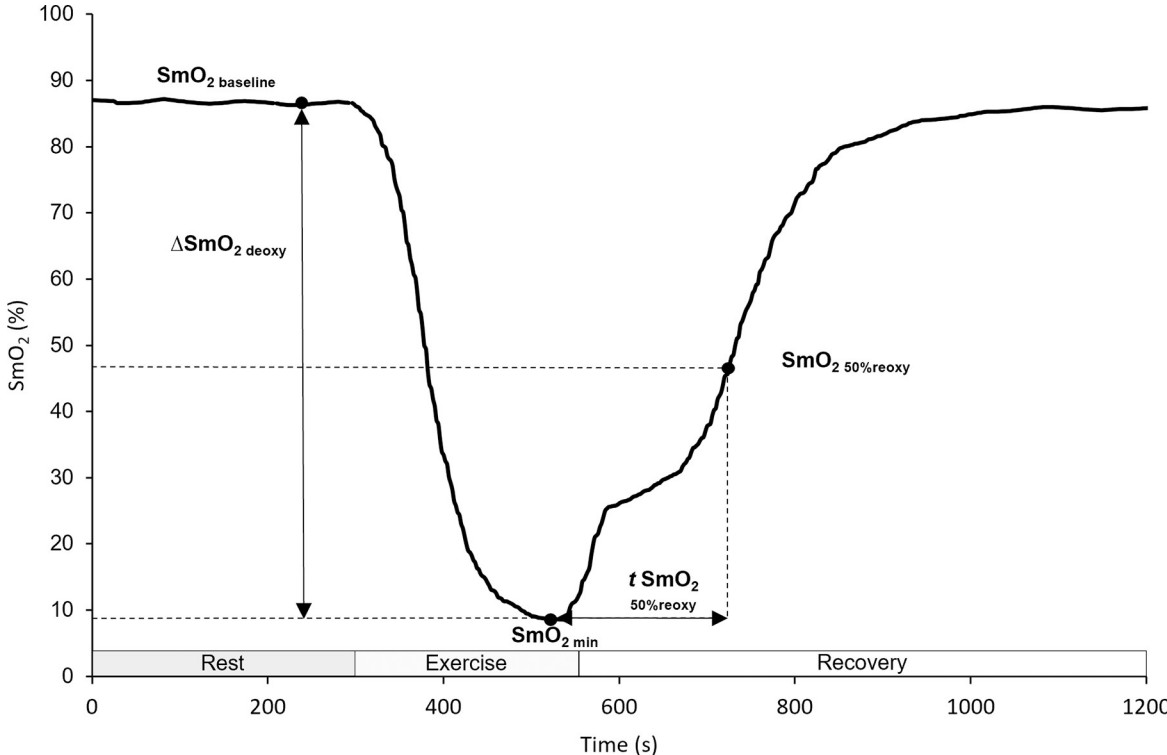

**Fig 2. Representative example of SmO$_2$-derived parameters, based on information from the author.** $SmO_{2\ baseline}$, baseline of muscle oxygen saturation; $\Delta SmO_{2\ deoxy}$, amplitude of muscle oxygen deoxygenation; $SmO_{2\ min}$, minimum of muscle oxygen saturation; $t\ SmO2_{50\%reoxy}$, time to recover 50% of muscle oxygen saturation; $SmO_{2\ 50\%reoxy}$, 50% of muscle oxygen reoxygenation.

**Perceived exertion (RPE).** The 15-point Rate of Perceived Exertion 6–20 (Borg RPE scale 6–20) Portuguese version was used to determine the perceived exertion [24]. Participants were asked to report an overall ($RPE_{ove}$) and local lower limbs ($RPE_{mus}$) perceived exertion after each set. A rating of 6 was correspondent with *no exertion at all* and a rating of 20 was correspondent with *maximal exertion*.

## Statistical analysis

The Shapiro-Wilk test was used to test the distribution of the data. Paired samples t-test and related samples Wilcoxon signed-rank test (the corresponding non-parametric test) were used to compare the outcomes between both testing protocols. The effect of type of muscle contraction over the 3 sets was assessed by repeated measures ANOVA and Friedman test (the equivalent non-parametric test), followed by the Bonferroni post hoc test to identify significant differences between each pairwise ($p \leq 0.05$). Effect size (ES) values of $\leq 0.2$, between 0.21, and 0.8, and $> 0.8$ were classified as small, moderate, and large, respectively, in paired samples t-test, repeated measures ANOVA and Friedman test [25], and in Wilcoxon signed-rank test, ES values of $\leq 0.147$, between 0.147, and 0.330, between 0.330 and 0.474, and $\geq 0.474$ were classified as negligible, small, medium, and large, respectively [26]. Analyses were performed using SPSS software V27.0 (IBM SPSS Statistics for Windows, Armonk, NY: IBM Corp.) and the results were presented as means ± standard deviation (SD) when they presented normal distribution, or median (25th - 75th percentiles) when those assumptions failed.

## Results

### $SmO_2$ responses between muscle contraction types and between the 3 sets

The $SmO_{2\,avg}$, $SmO_{2\,min}$, $\Delta SmO_{2\,deoxy}$ and $t\,SmO_{2\,50\%reoxy}$ values in response to both types of muscle contractions are shown in Fig 3.

In DYN, the SL muscle $SmO_{2\,avg}$ (54.5 ± 18.3%) was lower compared to the ISO (67.4 ± 18.0%) in 1st set, $t(9) = -4.342$, $p = 0.002$, $ES = -1.37$, and in the 2nd set (55.2 ± 19.0% vs. 62.8 ± 18.2%), $t(9) = -2.341$, $p = 0.044$, $ES = -0.74$. No differences were observed on $SmO_{2\,avg}$ in the VL, LG and ST muscles between protocols. No significant differences were seen between sets in each protocol.

The SL muscle $SmO_{2\,min}$ was lower in DYN when compared to the ISO in the 1st set (31.3 ± 11.8% vs. 43.1 ± 9.4%), $t(9) = -2.563$, $p = 0.031$, $ES = -0.81$, in the 2nd set (27.6 ± 11.5% vs. 40.9 ± 15.1%), $t(9) = -3.786$, $p = 0.004$, $ES = -1.20$, and in the 3rd (27.5 ± 13.3% vs. 41.1 ± 14.8%), $t(9) = -3.423$, $p = 0.008$, $ES = -1.08$. It means that the DYN promotes lower values in this muscle. No differences were observed on $SmO_{2\,min}$ in the VL, LG and ST muscles, neither between sets.

In the $\Delta SmO_{2\,deoxy}$, the DYN presented lower values compared to the ISO in all sets in SL muscle: 1st set, 32.6 ± 25.4% vs. 20.6 ± 15.2%, $t(8) = 2.995$, $p = 0.017$, $ES = 1.00$; 2nd set, 31.9 ± 25.1% vs. 13.2 ± 6.2%, $t(8) = 2.947$, $p = 0.019$, $ES = 0.98$; and 3rd, 34.6 ± 25.8% vs. 13.5 ± 7.7%, $t(8) = 2.412$, $p = 0.042$, $ES = 0.80$. No differences were observed in the other muscles or between sets.

The $t\,SmO_{2\,50\%reoxy}$ in VL of the ISO was higher compared with the DYN in the 3rd set (34.2 ± 14.4 s vs. 26.1 ± 12.2 s), $t(9) = -2.565$, $p = 0.030$, $ES = -0.81$. No differences were identified in the other muscles. No differences were observed in $tHb_{avg}$ and $tHb_{min}$ between the two exercise modes.

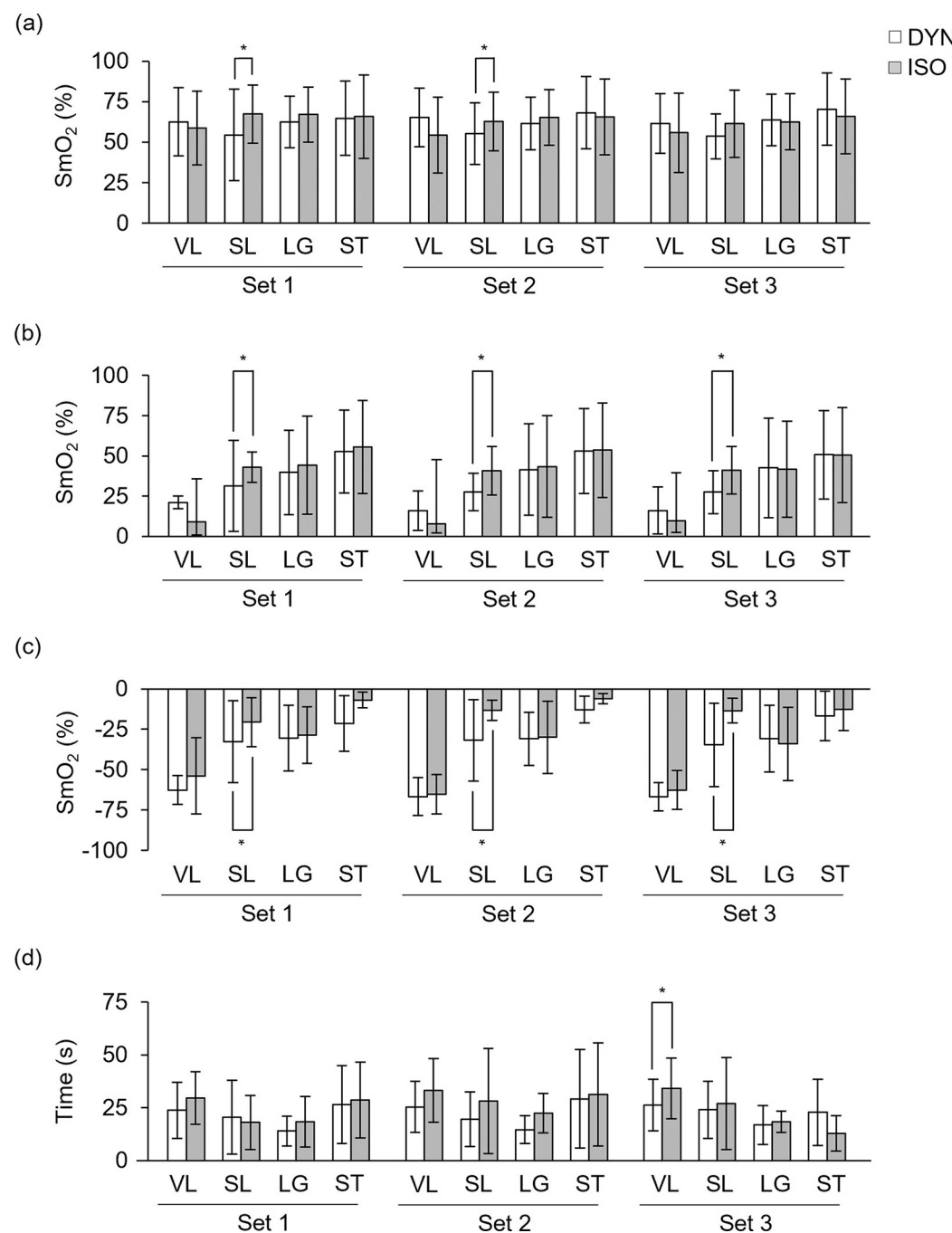

**Fig 3.** $SmO_{2\ avg}$ (a), $SmO_{2\ min}$ (b), $\Delta SmO_{2\ deoxy}$ (c) and $t\ SmO_{2\ 50\%reoxy}$ (d) results. DYN, dynamic contraction protocol; ISO, isometric contraction protocol; VL, *vastus lateralis*; SL, *soleus*; LG, *longissimus*; ST, *semitendinosus*. * $p \leq 0.05$.

## Cardiovascular, haemodynamic, and subjective responses to back squat exercise

Exercise and post-exercise HR, MAP, RPP, $RPE_{ove}$, and $RPE_{mus}$ main findings are shown in Table 2.

**Table 2. Cardiovascular, haemodynamic and perceived exertion responses to dynamic contraction protocol (DYN) and isometric contraction protocol (ISO).**

|  | 1st set | | 2nd set | | 3rd set | |
|---|---|---|---|---|---|---|
|  | DYN | ISO | DYN | ISO | DYN | ISO |
| HR (bpm) | 120.3 ± 15.9 | 115.9 ± 12.6 | 120.4 ± 15.1 | 114.5 ± 12.7 | 119.6(114.0–130.7)[#$] | 107.8(105.8–125.5)* |
| MAP (mmHg) | 99.0 ± 10.7 | 96.1 ± 8.7 | 99.7 ± 9.2 | 93.0 ± 3.4* | 99.1 ± 8.2 | 93.9 ± 8.9* |
| RPP (mmHg·bpm) | 14026.0 ± 2396.2 | 13634.5 ± 2147.7 | 1583.7 ± 2764.1[#] | 13869.0 ± 2579.3* | 15606.7 ± 3410.7 | 14534.7 ± 1819.4 |
| RPE$_{ove}$ (a.u.) | 13.5 ± 2.2 | 13.7 ± 2.2 | 13.0(13.3–15.0) | 15.0(13.3–16.0) | 14.2 ± 1.9[#] | 14.4 ± 2.1 |
| RPE$_{mus}$ (a.u.) | 13.0(13.0–15.8) | 15.1(14.3–16.8) | 14.5 ± 2.1 | 15.3 ± 2.3 | 14.0(13.3–14.8) | 16.0(15.3–17.5)* |

The values are mean ± standard deviation and median (25th - 75th percentiles). HR, heart rate; MAP, mean arterial pressure; RPP, rate pressure product; RPE$_{ove}$, overall perceived exertion; RPE$_{mus}$, muscular perceived exertion; * different from DYN ($p \leq 0.05$); # different from 1st set ($p \leq 0.05$); $ different from 2nd set ($p \leq 0.05$).

In DYN, HR was significantly higher when compared to ISO during the 3rd set, Z = -2.397, $p = 0.017$, ES = 0.86. There was a statistically significant effect of sets on HR in DYN, $F(2,9) = 4.88$, $p = 0.041$, $\eta^2 = 0.011$, presenting lower values on the 1st set in relation to the 3rd set (120.3 ± 15.9 bpm vs. 123.6 ± 16.2 bpm, $p = 0.048$) and on the 2nd set in relation to the 3rd set (120.4 ± 15.1 bpm vs. 123.6 ± 16.2 bpm, $p = 0.09$). MAP was significantly higher in DYN compared to ISO in the 2nd set, $t(8) = 3.195$, $p = 0.013$, ES = 1.07 and in the 3rd set, $t(8) = 2.909$, $p = 0.020$, ES = 0.97. No significant differences were found between sets in each protocol. MAP is related to the overall perfusion pressure. RPP was significantly higher in DYN when compared to ISO in the 2nd, $t(8) = 2.468$, $p = 0.039$, ES = 0.82. There was significant main effect of sets on RPP in DYN, $F(2,8) = 5.36$, $p = 0.016$, $\eta^2 = 0.063$, exhibiting lower values on the 1st set compared to the 2nd set (14026.0 ± 2396.2 mmHg · bpm vs. 1583.7 ± 2764.1 mmHg · bpm, $p = 0.002$). Although no significant differences were observed in RPE$_{ove}$ between protocols, there was a statistically significant effect of sets in DYN, $\chi^2(2) = 6.65$, $p = 0.036$, presenting lower values on the 1st set in relation to the 3rd set (13.5 [12.0–14.8] vs. 14.0 [13.0–15.0] a.u., $p = 0.009$). RPE$_{mus}$ was significantly higher in ISO compared to DYN in the 3rd, Z = -2.388, $p = 0.017$, ES = -1.00. No significant differences were seen between sets in each protocol. Still, the ISO showed a trend of higher RPE values compared to the DYN, in all sets.

## Discussion

This study provided preliminary data and evidence about the effect of the type of contraction on SmO$_2$-derived parameters, through a 3 sets back squat strength protocol, in four muscle groups simultaneously: VL, SL, ST, and LG. Cardiovascular, haemodynamic, and subjective responses were also compared between the dynamic and isometric contractions and between the sets. The main findings of the present study were that (i) SmO$_2$ $_{avg}$ showed significantly lower values during DYN compared to ISO in the SL muscle during the 1st and the 2nd set; (ii) SmO$_2$ $_{min}$ was significantly lower during DYN compared to ISO in SL muscle in all sets; (iii) ΔSmO$_2$ $_{deoxy}$ presented also significant differences in all sets, being higher in DYN compared to ISO; (iv) $t$ SmO$_2$ $_{50\%reoxy}$ after ISO was significantly longer compared to DYN in the VL muscle after the 3rd set; (v) in cardiovascular and haemodynamic parameters, i.e., HR and MAP, the DYN induced higher values in comparison to the ISO in the 3rd set and in MAP it was also in the 2nd set, while in RPP was in the 2nd set; and (vi) RPE$_{mus}$, was higher in ISO vs. DYN in every set.

At a physiological level, the use of different contraction modes can induce distinct SmO$_2$-derived parameters behaviour. Whereas a dynamic contraction is characterised by contraction-relaxation cycles, with blood flow being affected in the contraction phase and increasing in the relaxation phase, isometric contraction induces a constant intramuscular pressure [27–30]

which, depending on the load magnitude, may partially or totally restrict blood flow. Taking this into account and that the contribution of these 4 muscles to the back squat performance is different, so do the $SmO_2$ response, which reflects the balance between oxygen delivery and oxygen demand [17]. Within isometric contraction, the two existing forms can manifest different cardiovascular and muscular responses. For example, holding muscle action induces higher mean arterial pressure when compared with pushing muscle action [31]. The amplitude of variation of the mechanical muscular oscillations seems to be greater during holding muscle action in relation to pushing muscle action in muscles that present stabilizing function [32] and in prime movers it is the inverse.

The $SmO_{2\ avg}$ only showed differences in the SL muscle when contraction type is compared. These results may be due to innumerable factors, highlighting inter-individual muscle recruitment, which could only be accessed through electromyography (a method not used in our study). As in previous studies [33], these differences were observed in the first sets, probably due to a hyperemic response and an overshoot of $SmO_2$ factors that influence the response in subsequent sets [34].

Regarding $SmO_{2\ min}$, the SL muscle presented lower values in DYN, suggesting that this muscle is sensitive to contraction types in this variable during the back squat. The increase in intramuscular pressure which occurs at the expense of dynamic contractions can reduce blood flow, leading to a state of transient muscular hypoxia [35]. This reduction in blood flow which decreases the transport of oxygen to the muscle, in accordance with the greater energy expenditure by dynamic contractions in relation to isometric contractions with an equivalent load [36], produced lower minimum values. Although it was not the focus of our investigation, the SL muscle was the only one to show statistically significant differences and this may be due to several factors. The SL muscle, in relation to the VL and ST muscles, has higher penation angles [37], and a higher penation angle will increase the intramuscular pressure, decreasing the blood flow and consequently decreasing the value of $SmO_2$. On the other hand, the muscle fibers recruitment also seems to affect the $SmO_2$, with the recruitment of type I fibers reaching lower values of $SmO_2$ in relation to the recruitment of the other muscle fibers [20]. Since the SL muscle is the one with the highest number of type I fibers [38–41] this may be another explanation for the difference between muscles, however, only speculative.

The $\Delta SmO_{2\ deoxy}$ showed higher values in DYN, due to the fact that this variable it is closely related to the $SmO_{2\ min}$ value. Some authors argue that this variable, in line with others, i.e., HR, and blood lactate, can provide additional information regarding improvements in athlete performance [41–45]. This means that after an intervention program, the increase in deoxygenation is a favourable indicator. Only two studies compared this variable with the increment of dynamic and isometric contractions, using an equivalent load, and one of the studies did not show differences [36] while the other did [46], the latter corroborated by the results obtained in this study.

The calculation of the recovery time from $SmO_2$ can provide valuable information to define the interval time until the exercise is started again. This is because, although not evaluated directly, the phosphocreatine system has been related to $SmO_2$, both at the level of depletion and re-synthesis [34, 47]. In the 3rd set, the VL muscle showed a recovery time up to 50% of the baseline value which was significantly longer after the implementation of ISO. In training and performance context, a longer recovery time for VL and ST with dynamic contractions, and for SL and LG with isometric contractions could be usefull, until the beginning of the next sets.

Regarding tHb, it is important to know that it is not a valid indicator for assessing blood flow and its interpretation should be done carefully [12]. In a previous study [46], no significant differences were also observed between types of muscle contraction.

Concerning cardiovascular responses, namely, HR, there were higher values with the implementation of the DYN, as well as described elsewhere [46]. The higher values of HR evidenced during dynamic contractions may be explained by the higher cost of muscle activation and energy requirements [48, 49]. The different blood flow patterns between modes of contractions, the higher oxygen consumption, and the lower peripheral resistance with dynamic contractions are some of the factors that influence blood pressure response [50] presenting higher values in dynamic contractions in our study. RPP, also known as double product, was lower during ISO in consonance with previous studies [46] implying less myocardial effort and oxygen consumption.

The Borg scale was created to fill the existing gap of nonlinearity between perceptual ratings and both heart rate and power output observed with the 21-graded scale [51]. Even though there is a linear relationship, mainly in cardiovascular exercises, between RPE and heart rate using the 15-graded scale, and the prediction of heart rate from this scale prediction is facilitated (HR = RPE × 10), the response may be different depending on numerous factors. The subjective perceived exertion was assessed in two ways: overall and muscular. The $RPE_{ove}$ is one of the most common methods for monitoring the intensity and is related to feelings that are simple and easy to comprehend for the majority of individuals. Furthermore, $RPE_{mus}$ [52], which provides additional and specific information on the muscle groups that are having the most intervention. Both $RPE_{ove}$ and $RPE_{mus}$ showed higher values during the execution of isometric contractions, being significant in the $RPE_{mus}$. A possible explanation may be the effect of changes in blood flow on external perceptions. With a change of the blood flow, the intensity of the external perceptions of the individual intensifies, when compared to without occlusion [53, 54], and since during an isometric contraction this process occurs continuously, the participants may have felt a higher effort. Other factors that were not effectively analysed in this study, and that can influence the response in the RPE are exercise motivation [55], mental references [56], sensory experience [57], and comfort [58].

The study presents some limitations that cannot be dismissed. One of them is the non-use of an instrument that could equalize the workload between the two contraction types (i.e., strain gauge sensor, digital force transducer). Although both protocols were performed with the same exercise duration, interval rest between sets, and load lifted (kg), the 50% of 1RM represent a different relative intensity in the isometric protocol and the responses for the chosen angle of the isometric contraction cannot be extended to other angles.

The interpretation and practical translation of the data collected from the NIRS portable device is apparently the biggest challenge when this type of technology is applied, most probably because it is still a relatively new area. The present study highlights the advantage of monitoring in real-time the $SmO_2$-derived parameters together with other physiological variables, assuming a preponderant role in what is a localized muscular effort in exercise and recovery, particularly in the recovery time between sets.

## Conclusions

The findings of this study demonstrate that regardless of the type of contraction, the back squat exercise at 50% of 1RM does not seem to promote great changes in $SmO_2$ in the studied muscles, except for *soleus* muscle when referring to the minimum value and the amplitude of deoxygenation reached in exercise. In fact, the biggest changes seem to be related to cardiovascular parameters, having a more accentuated alteration with the imposition of dynamic contractions. On the other hand, perceived exertion responses to exercise were higher in isometric contractions. This may be an interesting aspect regarding the training load monitoring, as the load perception was higher with the isometric contractions, but effectively it was the dynamic contractions that had the greatest effect on the studied variables.

## Acknowledgments

The authors would like to thank the participants who gave up their time to participate in this study.

## Author Contributions

**Conceptualization:** Daniel Santarém, Catarina Abrantes.

**Data curation:** Daniel Santarém, Isabel Machado, Jaime Sampaio, Catarina Abrantes.

**Formal analysis:** Daniel Santarém, Isabel Machado, Jaime Sampaio, Catarina Abrantes.

**Investigation:** Daniel Santarém, Isabel Machado, Catarina Abrantes.

**Methodology:** Daniel Santarém, Isabel Machado, Catarina Abrantes.

**Project administration:** Catarina Abrantes.

**Resources:** Jaime Sampaio, Catarina Abrantes.

**Software:** Daniel Santarém.

**Supervision:** Jaime Sampaio, Catarina Abrantes.

**Validation:** Isabel Machado, Jaime Sampaio, Catarina Abrantes.

**Visualization:** Daniel Santarém, Isabel Machado, Jaime Sampaio, Catarina Abrantes.

**Writing – original draft:** Daniel Santarém.

**Writing – review & editing:** Isabel Machado, Jaime Sampaio, Catarina Abrantes.

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
