## [Decision Letter · Decision Letter 0]

5 Dec 2022

PONE-D-22-26930The effect of dynamic and isometric contraction type on cardiovascular, perceptual and near-infrared spectroscopy parameters: A pilot studyPLOS ONE

Dear Dr. Santarém,

Thank you for submitting your manuscript to PLOS ONE. After careful consideration, we feel that it has merit but does not fully meet PLOS ONE’s publication criteria as it currently stands. Therefore, we invite you to submit a revised version of the manuscript that addresses the points raised during the review process.

 Please submit your revised manuscript by Jan 19 2023 11:59PM. If you will need more time than this to complete your revisions, please reply to this message or contact the journal office at plosone@plos.org. Please include the following items when submitting your revised manuscript:A rebuttal letter that responds to each point raised by the academic editor and reviewer(s). You should upload this letter as a separate file labeled 'Response to Reviewers'.A marked-up copy of your manuscript that highlights changes made to the original version. You should upload this as a separate file labeled 'Revised Manuscript with Track Changes'.An unmarked version of your revised paper without tracked changes. You should upload this as a separate file labeled 'Manuscript'.

We look forward to receiving your revised manuscript.

Kind regards,

Emiliano Cè

Academic Editor

PLOS ONE

Journal Requirements:

Additional Editor Comments:

Dear Authors, 

one expert in the field reviewed your manuscript finding some methodological issues you should consider while revising the article.

Reviewers' comments:

Reviewer's Responses to Questions

**Comments to the Author**

1. Is the manuscript technically sound, and do the data support the conclusions?

Reviewer #1: Yes

2. Has the statistical analysis been performed appropriately and rigorously? 

Reviewer #1: Yes

3. Have the authors made all data underlying the findings in their manuscript fully available?

Reviewer #1: Yes

4. Is the manuscript presented in an intelligible fashion and written in standard English?

Reviewer #1: Yes

5. Review Comments to the Author

Reviewer #1: General:

The writing is generally excellent, and the study is interesting and unique.

Of course, the most obvious issue with the paper is the sample size of 7. Even if this is a ‘pilot’, I often suggest to authors to simply ‘keep going’ and collect more data. The introduction and methods section would not have to change, and depending on the ‘new’ results, the discussion may only change slightly. Since this is not a training/longitudinal study, I see little reason why the study cannot/should not be continued until a more suitable sample size is achieved. If for whatever reason, this is not possible, then the authors should clearly state why collection was stopped after only 7 participants.

Title:

The title could be clearer. At present, readers may be unsure if you are testing different kinds of isometric contraction (holding vs pushing, for example. I suggest the title be changes to something like:

“A comparison of dynamic and holding isometric contractions on cardiovascular, perceptual and near-infrared spectroscopy parameters: A pilot study”

Abstract:

The abstract is well written, and makes me want to read more.

The first thing that comes to mind is that it seems odd that sig differences were only seen for select measure and sets, instead of after each set. This makes me think that finding were potentially random chance, which is of course an issue with small samples/’pilot’ studies. I hope the authors address these questions in the body of the article.

Introduction:

Line 32: remove the period before reference 1.

Lines 32-33: I do not believe that the semitendinosus is a prime mover, but is more of a synergist. Feel free to argue the point, but perhaps move the semitendinosus with the other muscles as a synergist, or stabilizer etc.

Paragraph 2 is excellent. However, it would be valuable to briefly mention the difference between holding and pushing isometrics, and how holding are particularly understudies, despite arguably being more practical/easy to do in a typical weightroom setting.

Paragraph 3 is to technical, and brings up too many points that are not particularly relevant to the study/paper as a whole (i.e., melanin). I suggest simplifying this paragraph, and sticking to the points that are critical, such as deoxy, and re-oxy, and why those might be things practitioners may care about.

I also suggest making the purpose statement, and hypothesis its own small paragraph.

Methods:

The methods seem good, though to be honest, MOXY and other tools like that are far from my expertise. I hope the other reviewer(s) is knowledgeable in this regard.

Good work using the Bonferroni post hoc. I often see underpowered studies skipping the correction at all, or using more lenient corrections such as Tukey’s etc.

It should be clear why/when means vs medians were used for reporting. Did this have to do with distribution/the Shapiro-Wilk results?

Results:

Remove the spaces between numbers and the ‘%’ sign. I.e., ‘47.7 %’ should be ’47.7%’.

Some interesting results, esp the perceived exertion and HR going opposite directions between the two conditions. Since the SL is not a prime mover, perhaps relating back to some of the literature comparing pushing vs holding isometrics could be valuable to the reader. Ie., scientist such as Schaefer and Bittmann, and Roger Enoka’s group have generally found more activation and/or hemodynamic response in the supporting/synergist muscles during holding contractions, whereas the prime mover is more active/affected in the pushing isometrics.

Discussion:

Generally I wish the authors tried to better explain why some of the response differences were only seen in one set.

First and second paragraphs can be combined.

The small paragraph on lines 252-255 can be combined with the paragraph above it. The authors make good points about needing EMG etc. to further uncover the findings.

Generally try to avoid paragraphs that are 2 sentences or less, and try to incorporate them into other paragraphs.

While I understand that is difficult to understand fully, it would be interesting for the authors to try and explain why the ISO resulted in higher RPE, whereas the dynamic resulted in higher HR; esp since the 6-20 RPE scale was created to corelate with heartrate (60-200 BPM). Anecdotally/in my experience, participants tend to feel almost ‘board’, or even believe they are not doing anything useful during ISO contractions, and therefore may feel less motivated. Where it is clear they are ‘accomplishing’ something during DYN contractions. This may play into perceptions.

The most important limitation is missing. SAMPLE SIZE. Please be VERY clear about this, even so, I highly recommend continuing this study until the sample is into the double digits.

Figures/Tables:

Nice, no need for change or additional figures or tables.

6. PLOS authors have the option to publish the peer review history of their article (what does this mean?). If published, this will include your full peer review and any attached files.

Reviewer #1: **Yes: **Dustin J Oranchuk

---

## [Author Response · Author response to Decision Letter 0]

19 Jan 2023

Reviewer #1: General:

The writing is generally excellent, and the study is interesting and unique.

Of course, the most obvious issue with the paper is the sample size of 7. Even if this is a ‘pilot’, I often suggest to authors to simply ‘keep going’ and collect more data. The introduction and methods section would not have to change, and depending on the ‘new’ results, the discussion may only change slightly. Since this is not a training/longitudinal study, I see little reason why the study cannot/should not be continued until a more suitable sample size is achieved. If for whatever reason, this is not possible, then the authors should clearly state why collection was stopped after only 7 participants.

 - The sample size was increased to 10 participants.

Title:

The title could be clearer. At present, readers may be unsure if you are testing different kinds of isometric contraction (holding vs pushing, for example. I suggest the title be changes to something like:

“A comparison of dynamic and holding isometric contractions on cardiovascular, perceptual and near-infrared spectroscopy parameters: A pilot study”

 - The title was improved and is clearer now. “Comparing the effects of dynamic and holding isometric contractions on cardiovascular, perceptual, and near-infrared spectroscopy parameters: A pilot study”. Please see lines 1-4. 

Abstract:

The abstract is well written, and makes me want to read more.

The first thing that comes to mind is that it seems odd that sig differences were only seen for select measure and sets, instead of after each set. This makes me think that finding were potentially random chance, which is of course an issue with small samples/’pilot’ studies. I hope the authors address these questions in the body of the article.

 - We have edited the text accordingly. The concept of holding was added to the text, as well as some minor changes in the sample description data and statistical results. Please see the Abstract.

Introduction:

Line 32: remove the period before reference 1.

 - We have edited the text accordingly. Please see line 33.

Lines 32-33: I do not believe that the semitendinosus is a prime mover, but is more of a synergist. Feel free to argue the point, but perhaps move the semitendinosus with the other muscles as a synergist, or stabilizer etc.

 - We have edited the text accordingly. During this exercise, the vastus lateralis (VL) muscle act as primary mover, longissimus (LG) and semitendinosus (ST) muscles act as stabilizers, and soleus (SL) muscle act as secondary capacity. Please see lines 33-35.

Paragraph 2 is excellent. However, it would be valuable to briefly mention the difference between holding and pushing isometrics, and how holding are particularly understudies, despite arguably being more practical/easy to do in a typical weightroom setting.

 - We have edited the text accordingly and we had to add new references related to the characterisation of the different modes of isometric contractions. In fact, we completely agree that the information clarifying what type of isometric contraction is being studied is extremely relevant. Please see lines 40-45.

Paragraph 3 is to technical, and brings up too many points that are not particularly relevant to the study/paper as a whole (i.e., melanin). I suggest simplifying this paragraph, and sticking to the points that are critical, such as deoxy, and re-oxy, and why those might be things practitioners may care about.

 - We have edited the text accordingly. Some relevant information related to NIRS has been moved to the methodology chapter. Please see lines 59-61 and 81-82.

I also suggest making the purpose statement, and hypothesis its own small paragraph.

 - We have edited the text accordingly. Please see lines 66-71.

Methods:

The methods seem good, though to be honest, MOXY and other tools like that are far from my expertise. I hope the other reviewer(s) is knowledgeable in this regard.

Good work using the Bonferroni post hoc. I often see underpowered studies skipping the correction at all, or using more lenient corrections such as Tukey’s etc.

It should be clear why/when means vs medians were used for reporting. Did this have to do with distribution/the Shapiro-Wilk results?

 - We have edited the text accordingly. We could not agree more that it should be explained why averages and medians appear, since the averages are when the data has a normal distribution and the medians are when the data does not have a normal distribution. Please see lines 182-186 and 190-193.

Results:

Remove the spaces between numbers and the ‘%’ sign. I.e., ‘47.7 %’ should be ’47.7%’.

 - We have edited the text accordingly. Please see lines 204, 205, 208, 209, 210, 214, 215.

Some interesting results, esp the perceived exertion and HR going opposite directions between the two conditions. Since the SL is not a prime mover, perhaps relating back to some of the literature comparing pushing vs holding isometrics could be valuable to the reader. Ie., scientist such as Schaefer and Bittmann, and Roger Enoka’s group have generally found more activation and/or hemodynamic response in the supporting/synergist muscles during holding contractions, whereas the prime mover is more active/affected in the pushing isometrics. 

 - We have edited the text accordingly. Some differences between holding and pushing were emphasized. The amplitude of variation of the mechanical muscular oscillations seems to be greater during holding muscle action in relation to pushing muscle action in muscles that present stabilizing function and in prime movers it is the inverse. Please see lines 262-267.

Discussion:

Generally I wish the authors tried to better explain why some of the response differences were only seen in one set.

 - We have edited the text accordingly. According to the available literature, the physiological rational is related to hyperaemia in response to exercise. Please see lines 270-272.

First and second paragraphs can be combined.

 - We have edited the text accordingly. Please see line 244-254.

The small paragraph on lines 252-255 can be combined with the paragraph above it. The authors make good points about needing EMG etc. to further uncover the findings.

 - Since additional information has been included, we did not consider it necessary to add it to the paragraph above. We await feedback.

Generally try to avoid paragraphs that are 2 sentences or less, and try to incorporate them into other paragraphs.

 - We have edited the text accordingly. 

While I understand that is difficult to understand fully, it would be interesting for the authors to try and explain why the ISO resulted in higher RPE, whereas the dynamic resulted in higher HR; esp since the 6-20 RPE scale was created to corelate with heartrate (60-200 BPM). Anecdotally/in my experience, participants tend to feel almost ‘board’, or even believe they are not doing anything useful during ISO contractions, and therefore may feel less motivated. Where it is clear they are ‘accomplishing’ something during DYN contractions. This may play into perceptions.

 - We have edited the text accordingly. Effectively, changes in blood flow can affect external perceptions. Please see lines 312-317 and 321-326.

The most important limitation is missing. SAMPLE SIZE. Please be VERY clear about this, even so, I highly recommend continuing this study until the sample is into the double digits.

 - The sample size was increased to 10 participants.

Figures/Tables:

Nice, no need for change or additional figures or tables.

Additional information from the authors:

Abstract

 - The results, in descriptive and numerical terms, were rectified for all variables.

Material and methods | Participants

 - The results, in numerical terms, were rectified for all variables.

 - One information that was in the introduction was changed for this topic. Please see lines 81-82.

Results

 - The results, in descriptive and numerical terms, were rectified for all variables.

Discussion

 - The results, in descriptive terms, were rectified.

References

 - New references have been added, duly marked.

Figures

 - Figure 3 was elaborated again, changing only the results.

---

## [Decision Letter · Decision Letter 1]

2 Feb 2023

Comparing the effects of dynamic and holding isometric contractions on cardiovascular, perceptual, and near-infrared spectroscopy parameters: A pilot study

PONE-D-22-26930R1

Dear Dr. Santarém,

We’re pleased to inform you that your manuscript has been judged scientifically suitable for publication and will be formally accepted for publication once it meets all outstanding technical requirements.

Kind regards,

Emiliano Cè

Academic Editor

PLOS ONE

Additional Editor Comments (optional):

Reviewers' comments:

Reviewer's Responses to Questions

**Comments to the Author**

1. If the authors have adequately addressed your comments raised in a previous round of review and you feel that this manuscript is now acceptable for publication, you may indicate that here to bypass the “Comments to the Author” section, enter your conflict of interest statement in the “Confidential to Editor” section, and submit your "Accept" recommendation.

Reviewer #1: All comments have been addressed

2. Is the manuscript technically sound, and do the data support the conclusions?

Reviewer #1: Yes

3. Has the statistical analysis been performed appropriately and rigorously? 

Reviewer #1: Yes

4. Have the authors made all data underlying the findings in their manuscript fully available?

Reviewer #1: Yes

5. Is the manuscript presented in an intelligible fashion and written in standard English?

Reviewer #1: Yes

6. Review Comments to the Author

Reviewer #1: I am impressed with the alterations to the article! Esp the increase in sample size from 7 to 10, which was my major concern.

The text is more clear, and the paper is excellent.

I require no additional edits, and fully endorse publication.

Well done!

7. PLOS authors have the option to publish the peer review history of their article (what does this mean?). If published, this will include your full peer review and any attached files.

Reviewer #1: **Yes: **Dustin J Oranchuk

---

## [Editor Report · Acceptance letter]

8 Feb 2023

PONE-D-22-26930R1 

Comparing the effects of dynamic and holding isometric contractions on cardiovascular, perceptual, and near-infrared spectroscopy parameters: A pilot study 

Dear Dr. Santarém:

I'm pleased to inform you that your manuscript has been deemed suitable for publication in PLOS ONE. Congratulations! Your manuscript is now with our production department. 

Kind regards, 

on behalf of

Professor Emiliano Cè 

Academic Editor

PLOS ONE